# Towards Scale Balanced 6-DoF Grasp Detection in Cluttered Scenes

**Haoxiang Ma, Di Huang**[*]
State Key Laboratory of Software Development Enviroment
School of Computer Science and Engineering
Beihang University, Beijing, 10019, China
{mahaoxiang822, dhuang}@buaa.edu.cn

**Abstract:** In this paper, we focus on the problem of feature learning in the presence of scale imbalance for 6-DoF grasp detection and propose a novel approach to especially address the difficulty in dealing with small-scale samples. A Multi-scale Cylinder Grouping (MsCG) module is presented to enhance local geometry representation by combining multi-scale cylinder features and global context. Moreover, a Scale Balanced Learning (SBL) loss and an Object Balanced Sampling (OBS) strategy are designed, where SBL enlarges the gradients of the samples whose scales are in low frequency by apriori weights while OBS captures more points on small-scale objects with the help of an auxiliary segmentation network. They alleviate the influence of the uneven distribution of grasp scales in training and inference respectively. In addition, Noisy-clean Mix (NcM) data augmentation is introduced to facilitate training, aiming to bridge the domain gap between synthetic and raw scenes in an efficient way by generating more data which mix them into single ones at instance-level. Extensive experiments are conducted on the GraspNet-1Billion benchmark and competitive results are reached with significant gains on small-scale cases. Besides, the performance of real-world grasping highlights its generalization ability. Our code is available at https://github.com/mahaoxiang822/Scale-Balanced-Grasp.

**Keywords:** grasp detection, point-cloud representation, scale balance

## 1 Introduction

Grasping is a fundamental task in robotic manipulation, and grasp detection aims to generate rich gripper configurations on objects for stable lift. It is expected to deal with objects varying in shape, size, appearance, material, pose *etc*. Traditional methods usually work with 3D models of objects and policies are manually designed [1, 2, 3]. However, these methods highly rely on the accuracy of pose estimation and are not applicable to unseen objects, limiting their use in real-world scenarios. With the innovation of depth sensors and the development of deep learning, data-driven methods have been greatly advanced [4, 5, 6], offering the advantage in generalization. More recently, several studies have extended 6-DoF grasp detection to a more difficult case, investigating diverse grasping poses for objects in clutter [7, 8, 9]. Although such attempts deliver promising results, they incline to large- and medium-scale objects or parts. There exist quite a number of hard examples. As shown in Fig. 1 (a), there are many qualified grasps detected on the large red box in (1), but very few for the bolt in (2) and the fork in (3). For the bottle in (4), there exist several valid grasps on the body while none is on the red head.

Indeed, grasping on small objects (*e.g.* the bolt in (2) and the fork in (3) in Fig. 1 (a)) or small parts of large- and medium-scale objects (*e.g.* the red head of the bottle in (4) in Fig. 1 (a)) is still in low precision and recall. Here, we formally define the scale of a grasp as the minimal width between two fingers of the gripper, which is determined by the pose of the grasp and the shape of the region. To illustrate the correlation between the grasp scale and quality, we visualize the scale distribution of the objects and their corresponding grasp performance by the baseline method on the GraspNet-

---

[*]Corresponding author.

6th Conference on Robot Learning (CoRL 2022), Auckland, New Zealand.

1Billion benchmark [8]. As depicted in Fig. 1 (b), current grasp detection method performs well on the objects which seldom have small-scale ground truth grasps. On the contrary, for the objects with a large proportion of small-scale ground truth grasps, the baseline method tends to fail.

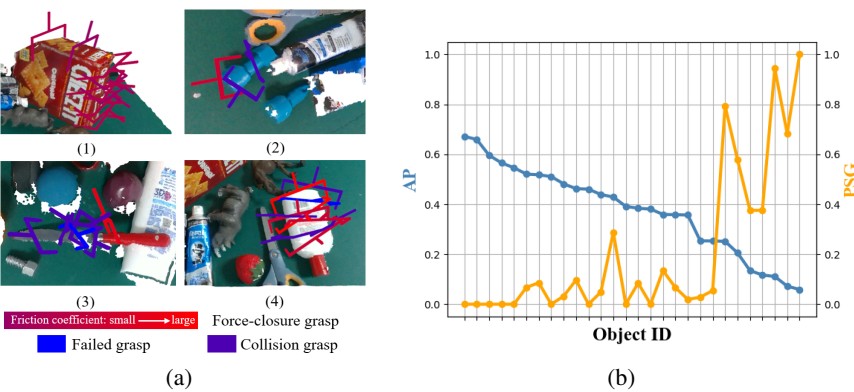

Figure 1: (a) Grasping objects of varying scales by GraspNet [8]. (b) Grasping Average Precision (AP) and Percentage of Small-scale Grasps (PSG) for different objects.

It is really challenging to improve the results of such hard cases towards scale-balanced grasp detection. **Firstly**, for a local region, the ambiguity of its geometry increases when its size becomes smaller since information conveyed reduces. [8, 10, 11] extract local geometry features through specific modules, but due to fixed receptive fields, they are not so competent to distinguish similar but different shapes. **Secondly**, the scale distribution of grasps for daily objects in cluttered scenes is imbalanced. Large- and medium-scale grasps dominate and thus suppress the learning on small-scale ones. To the best of our knowledge, this issue has not been investigated in 6-DoF grasp detection before. **Thirdly**, the noise of mainstream depth sensors (*e.g.* Intel RealSense and MS Kinect) is serious, which degrades shape description, in particular for small regions. [12, 13] provide tentative solutions. Unfortunately, the former applies point completion, mainly working on seen objects because completion of missing regions of unknown objects is confusing without prior knowledge [14], while the latter conducts denoising, which requires rendering raw data by adding noise of different patterns to clean point-clouds for training models, struggling in generalizing to the complicated real-world noise from RGB-D sensors.

This paper proposes a novel approach to 6-DoF grasp detection, which addresses the difficulties aforementioned in a unified framework. Specifically, a Multi-scale Cylinder Grouping (MsCG) module is introduced, where multi-scale cylinder features are combined to a global feature with additional context embedded for more comprehensive description of local shapes. Furthermore, a Scale Balanced Learning (SBL) loss and an Object Balanced Sampling (OBS) strategy are designed for the training and inference phases respectively. SBL enlarges the gradients of the samples whose scales are in low frequency (*i.e.* small) by apriori weights while OBS captures more points on small-scale objects with the help of an auxiliary segmentation network. They both alleviate the influence of the uneven distribution of grasp scales. Additionally, a data augmentation method, namely Noisy-clean Mix (NcM), is presented to facilitate training. It aims to bridge the domain gap between synthetic and raw scenes in an efficient way, by generating more data which mix them into single scenes at instance-level.

In summary, our contributions are as follows: (1) we introduce an MsCG module to enhance the representation of local shapes for grasp detection; (2) we design an SBL training loss and an OBS inference strategy to mitigate the imbalanced scale distribution of grasps in cluttered scenes; (3) we present NcM data augmentation by blending point-clouds with and without noise at instance-level to improve model training; and (4) we carry out extensive simulation and real-world experiments and reach competitive results with significant gains on small-scale grasps and a strong generalization ability.

## 2 Related Work

Grasping is a long-standing task in robotic manipulation. Recently, there has been a trend on 6-DoF grasp detection in cluttered scenes. [7] proposes a pioneering framework with point-cloud input, which first samples grasp candidates and then scores them with a Convolutional Neural Network (CNN). [9] employs a point-based neural network to score grasp candidates for improvements. Due to the fact that grasp candidate sampling is time-consuming and prone to objects (or parts) of thin shapes, a number of studies [15, 16, 8, 10, 17] handle this task in an end-to-end manner. [15] introduces a variational auto-encoder to generate grasp poses from point-clouds of single objects. To directly regress grasping in clutter, [16] builds a single-shot grasp proposal network. [8] generates diverse grasps for objects in clutter by constructing a discrete grasp pose space and predicting the gripper configurations progressively. For more accurate and robust grasps, [17] defines graspness as the quality metric to determine where to grasp and [10] focuses on capturing local features from points inside grasp regions. Although the methods above have made large progress, they generally work well on objects (or parts) of large- and medium-scales and have difficulty in tackling small-scale ones, leading to the imbalanced grasping detection performance.

As the overwhelming majority of the studies on 6-DoF grasp detection take point-clouds as input and such data often contain noise of depth sensors, there exists another line to ameliorate scene representation. [18, 19] bring additional clues to complement description where [18] uses multiple frames to map raw input to a Truncated Signed Distance Function (TSDF) to smooth noise and [19] integrates texture maps to deliver multi-modal features. Besides, [20] introduces implicit representations to model scenes and employs 3D reconstruction as an auxiliary task to refine data. Such solutions either require more information in inference or highly correlate to the specific model, suggesting the necessity of a plug and play alternative.

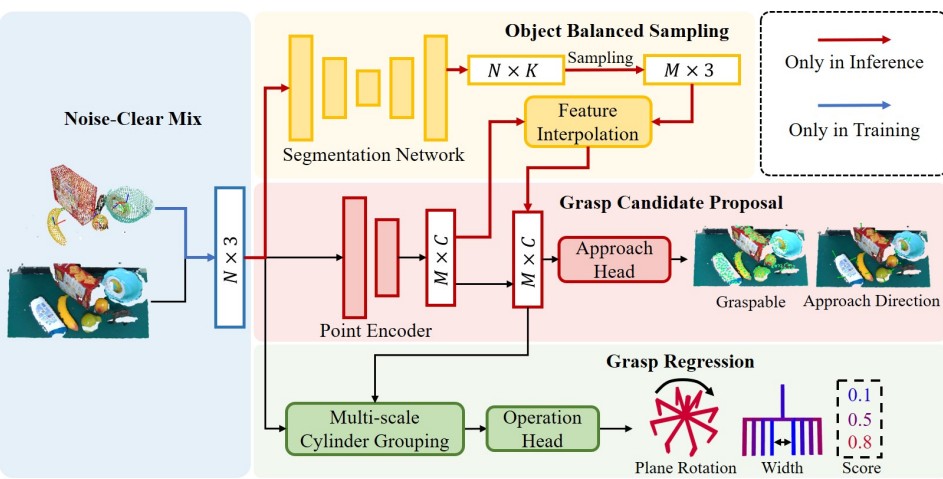

Figure 2: Pipeline of the proposed scale balanced grasp detection approach.

## 3 Methodology

The pipeline of the scale balanced grasp detection approach is demonstrated in Fig. 2. During training, we first conduct **Noisy-clean Mix** (NcM) augmentation, which synthesizes clean point-clouds from CAD models and mixes them with the raw noisy ones at instance-level. The new point-cloud $\in \mathbb{R}^{N \times 3}$ is employed as input, where $N$ is the number of points. After that, $M$ candidate points are sampled by Farthest Point Sampling (FPS) from the input and encoded as seed features $\in \mathbb{R}^{M \times C}$ by a transformer-based point encoder [21]. We predict whether these candidates can be grasped and the approach directions of positive candidates using an approach head. To regress other gripper configurations, **Multi-scale Cylinder Grouping** (MsCG) is built to extract local features for positive candidates from original point-clouds. An operation head is employed to predict the plane rotation, width and score of the grasp. During inference, **Object Balanced Sampling** (OBS) is used to explore more positive samples at different scales, where a 3D instance segmentation network is integrated to predict the 3D masks $\in (0, 1)^{N \times K}$ ($K$ is the number of instances) and new positions

$\in \mathbb{R}^{M \times 3}$ are evenly sampled in each mask. The seed features of these new samples are interpolated from the initial ones in scene and used for successive grasp configuration prediction.

## 3.1 Multi-scale Cylinder Grouping

Considering that grasps correspond to detailed local geometries, it is rather rough to choose a fixed Receptive Field (RF) to represent the grasp region. As Fig. 3 (a) displays, a single RF tends to incur ambiguity for the small-scale grasp in position A and miss crucial cues for the large-scale one in position B. To enhance geometric representations for shapes in different sizes, we propose the Multi-scale Cylinder Grouping (MsCG) module, which captures local features of a position with varying RFs and combines them to deliver a comprehensive description. Specifically, we set up several cylinders with different radii and crop related points. Radii are evenly distributed within the maximum width of the gripper. Multi-layer Perceptrons (MLPs) and max-pooling are employed to encode the points from different cylinders. To further improve the semantics, we introduce the corresponding seed features as guidance for local feature fusion at different scales, and a gate module is applied to screen information in them. Fig. 3 (b) shows the structure of the MsCG module. With the help of this module, our network achieves more accurate local geometric representations for grasps at different scales.

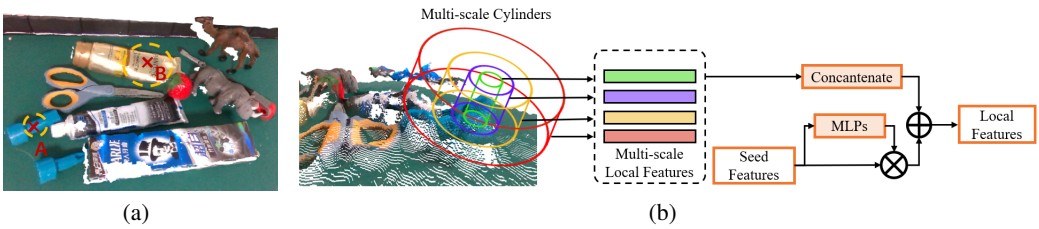

Figure 3: (a) Grasp regions at different scales for local feature extraction. (b) Structure of the Multi-scale Cylinder Grouping module.

## 3.2 Grasp Scale Balancing

Firstly, we give an analysis on the scale distribution of grasps in cluttered scenes on the training set of the GraspNet-1Billion benchmark [8]. 1,024 points are sampled by FPS in each scene and for each sample, we search for the grasp with the highest score in the $SO(3)$ space, whose gripper width is considered as the scale of the sample. Through this, for all scenes in the training set, we draw the grasp scale distribution, as shown in Fig. 4. It is observed that medium- and large-scale grasps are much more than small-scale ones. This imbalance issue degrades grasp detection in two aspects. First, in training, the model is prone to fit medium- and large-scale grasps whose percentage is high and thus suppress the learning of small-scale grasps. Second, in inference, few small-scale grasps can be sampled by current evenly sampling techniques in the 3D space such as FPS and voxel sampling. Simply enlarging the number of samples can indeed ease this problem, but the memory cost and run-time both sharply increase, making it impractical.

To address the imbalance issue in training, we design a Scale Balanced Learning (SBL) loss to weight the errors of the samples at different scales inspired by Cost-Sensitive Learning [22], where the weights are calculated based on the frequencies of the classes of samples. Different from the imbalance in the discrete classification problem, that on grasping scales is continuous and we thus divide the max width of the gripper into $T$ bins. For sample $i$ in the scene, its weight $W_i$ is calculated by:

$$W_i = 1 - \log \frac{C_i}{C_{max}} \tag{1}$$

where $C_i$ is the number of grasps of the same scale with sample $i$ and $C_{max}$ is the maximum of the $T$ scales. For negative samples where no successful grasp is in its $SO(3)$ space, we assign $W_n = 1$. With the loss in [8], we add the weight $W$ to the approach and operation heads:

$$Loss = W(Loss^{Approach} + \alpha Loss^{Rotation}) \tag{2}$$

where $\alpha$ is a trade-off hyper-parameter. Please refer to the supplementary material for more details.

To handle the imbalance issue in inference, we propose an Object Balanced Sampling (OBS) strategy, which adjusts the number of samples on different objects so that sufficient samples are obtained on them. In this case, we first train a 3D instance segmentation network in advance, *i.e.* a modified version of [23], only with point-clouds as input to generate masks for objects. We then uniformly carry out FPS on each foreground mask and each object has $\frac{M}{K}$ samples, where $M$ and $K$ are previously defined as the number of total samples and the number of instances respectively. The features of new sampled positions are linearly interpolated by three nearest neighbors from the original seed features in scene. With the OBS strategy, we are able to explore more grasp candidates on small-scale objects, benefiting small-scale grasp detection.

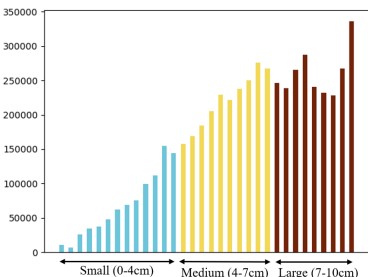

Figure 4: Grasp scale distribution on GraspNet-1Billion.

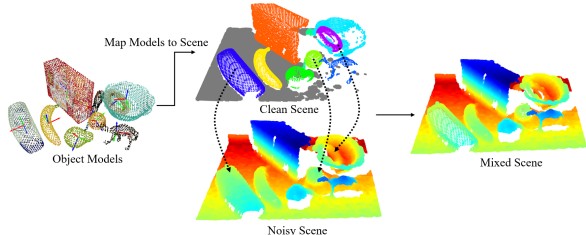

Figure 5: Pipeline of NcM data augmentation.

### 3.3 Noisy-clean Mix

The noise of point-clouds affects the accuracy of grasping detection, especially for small-scale cases. On the one hand, the noise destroys the geometry structure of the grasp, which makes it hard to detect. The attempts on point-cloud completion or denoising partially solve this problem during inference but numerous unseen objects and complicated noise patterns in the real-world prevent their further use. On the other hand, the noise degrades the training of grasp detection where the network struggles in learning the mapping from ambiguous shapes to grasp labels. Compared to larger geometries, the distortions due to noise on small ones are more serious, decreasing the contribution in training.

To overcome this downside, we propose a Noisy-clean Mix (NcM) data augmentation method. We introduce the clean version of the input point-clouds to increase the number of valid grasp samples. However, Sim-to-Real raises another challenge when only using synthetic clean point-clouds during training. Therefore, we mix noisy and clean data into single scenes, which reduces the domain gap between training and inference. Fig. 5 shows the process of NcM. With 3D models and poses of objects, we synthesize corresponding clean scenes by composing them to full point-clouds. We use the original point-clouds captured by the depth sensor as noisy scenes. For the missing parts in noisy scenes caused by camera poses and occlusions, we also remove the points in the same region of the clean point-clouds to keep the consistency. The mixed input is constructed by randomly replacing the objects in noisy scenes with the clean counterparts, leading to more valid samples to facilitate the learning of grasps.

## 4 Experiments

### 4.1 Benchmark, Baseline and Metric

We conduct all the simulation experiments on GraspNet-1Billion [8], which includes 190 cluttered scenes captured in the real-world by Realsense/Kinect cameras in 256 views. For each point on the object surface, grasps are annotated densely in its $SO(3)$ space by force closure estimation. It has nearly 1 billion annotations, offering rich grasps to explore the scale distribution in scene.

The baseline method is an enhanced version of GraspNet-baseline [8], where a transformer-based point encoder and a new graspable loss with a higher threshold are applied for better performance. For more details about the baseline, please refer to the supplementary material.

To evaluate the grasps detected in cluttered scenes, the precision of the top-$k$ ranked grasps is used. As in [8], $\mathbf{AP}_\mu$ is computed to represent the average *Precision@k* for $k$ ranging from 1 to 50 with friction $\mu$, and $\mathbf{AP}$ is obtained by the average of $\mathbf{AP}_\mu$, where $\mu$ varies from 0.2 to 1.2.

The original metric does not directly reflect the quality of grasps at different scales because the top ranked grasps mostly distribute on objects (or parts) with simple and large geometries. We thus give a new metric to better evaluate the scale-aware grasping quality in scene. The max width of the gripper (10cm in benchmark) is divided to three intervals, where widths in 0cm-4cm, 4cm-7cm, and 7cm-10cm are small-scale, medium-scale and large-scale. With such width masks, we evaluate $\mathbf{AP}_S$, $\mathbf{AP}_M$ and $\mathbf{AP}_L$ for grasps in small-, medium- and large-scale respectively.

## 4.2 Ablation Study

First, we validate the effectiveness of the proposed MsCG module, SBL loss, OBS strategy, and NcM augmentation method, and the results are listed in Table 1. All the studies are conducted on the real-world scenes captured by Intel RealSense D435. We evaluate $\mathbf{AP}_S$, $\mathbf{AP}_M$ and $\mathbf{AP}_L$, and calculate the mean to reflect the grasp quality at all the scales. According to Table 1, MsCG achieves significant improvements in all the metrics, which shows the importance of enhanced shape representations. For SBL, the performance of small-scale grasps is boosted by increasing the weights for such samples in training, comparable to that on medium- and large-scale grasps. NcM augmentation brings additional gains for almost all the settings, which demonstrates its necessity in reducing ambiguities during training. For fair validation of the advantage of OBS, we keep the same number of samples (1,024 in our experiment) and take Foreground Sampling (FS) as comparison, where we do FPS in foreground point-clouds. Compared to FS, OBS delivers an improvement of 2.37% for small-scale grasps in the seen set with stable results of medium- and large-scale grasps. We also note that OBS does not show an improvement on small-scale grasps in the novel set (but still comparable), and it is mainly caused by the inferior performance of the segmentation network for unseen objects.

| Model | Seen | | | | Similar | | | | Novel | | | |
|---|---|---|---|---|---|---|---|---|---|---|---|---|
| | $\mathbf{AP}_S$ | $\mathbf{AP}_M$ | $\mathbf{AP}_L$ | Mean | $\mathbf{AP}_S$ | $\mathbf{AP}_M$ | $\mathbf{AP}_L$ | Mean | $\mathbf{AP}_S$ | $\mathbf{AP}_M$ | $\mathbf{AP}_L$ | Mean |
| Baseline | 9.44 | 45.99 | 54.13 | 36.52 | 5.15 | 35.54 | 47.82 | 29.50 | 4.91 | 15.26 | 19.83 | 13.33 |
| Ours | 13.47 | 48.12 | 61.81 | **41.13** | 6.23 | 37.90 | 53.89 | **32.67** | 7.60 | 17.04 | 23.10 | **15.91** |
| w/o MsCG | 9.14 | 42.65 | 52.75 | 34.85 | 3.53 | 32.82 | 49.07 | 28.47 | 4.00 | 14.56 | 21.08 | 13.21 |
| w/o SBL | 10.50 | 47.20 | 63.26 | 40.32 | 4.98 | 37.02 | 54.52 | 32.17 | 5.97 | 18.03 | 23.12 | 15.71 |
| w/o NcM | 12.69 | 46.25 | 61.78 | 40.24 | 6.00 | 36.88 | 52.55 | 31.81 | 7.06 | 16.38 | 23.27 | 15.57 |
| Ours + FS | 15.92 | 52.73 | 64.56 | 44.40 | 8.78 | 42.67 | 57.01 | 36.15 | 9.42 | 18.64 | 24.23 | 17.43 |
| Ours + OBS | 18.29 | 52.60 | 64.34 | **45.08** | 10.03 | 42.77 | 57.09 | **36.63** | 9.29 | 18.74 | 24.36 | **17.46** |

Table 1: Ablation study of the proposed modules on scenes captured by Intel RealSense D435.

## 4.3 Analysis on Different Scales

We discuss how the proposed approach influences the number and success rate of grasps at different scales. For each object in scene, we select the top-10 ranked grasps for statistics and set the friction $\mu = 0.8$ for grasp successful checking. Results on seen objects are shown in Fig. 6.

For small-scale samples, MsCG, SBL, and OBS all increase the number and success rate of detected grasps. Note that NcM augmentation reduces the number of small-scale grasps but improves the success rate. By introducing more valid small-scale samples in learning, our network avoids to generate grasps in the region where the shape is seriously corrupted by noise. For medium- and large-scale samples, MsCG still shows a significant improvement in terms of the success rate.

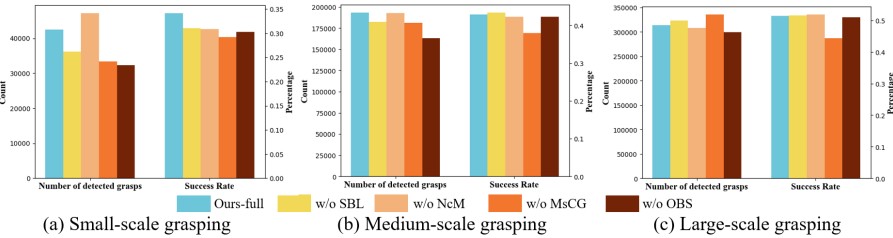

(a) Small-scale grasping     (b) Medium-scale grasping     (c) Large-scale grasping

Figure 6: The number and success rate of grasps at different scales on seen objects.

## 4.4 Visualization

We visualize some grasps generated by our approach on the objects that the baseline does not perform well. The results are shown in Fig. 7. Gripper poses in red are good grasps while those in purple and blue are collision and bad ones respectively. Objects in (1-3) are small-scale and our approach detects more successful grasps on them. The scissors in (4) and the bottle in (5) consist of several geometric parts in different sizes. Benefiting from the ability to detect grasps at various scales, our approach delivers diverse grasps for individual parts, helpful to the downstream manipulation task.

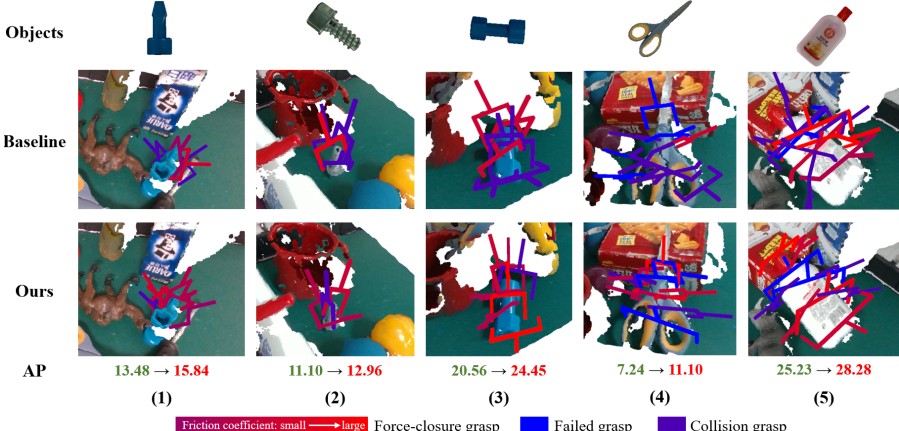

Figure 7: Visualization of detected grasps on hard objects.

## 4.5 Comparison with the State-of-the-art

We make fair comparison with the state-of-the-art counterparts on the data acquired by the RealSense camera, including [7, 6, 24, 9, 8, 19, 25, 17] and the results are displayed in Table 2.

| Model | Seen | | | Similar | | | Novel | | |
|---|---|---|---|---|---|---|---|---|---|
| | AP | $AP_{0.8}$ | $AP_{0.4}$ | AP | $AP_{0.8}$ | $AP_{0.4}$ | AP | $AP_{0.8}$ | $AP_{0.4}$ |
| GG-CNN [6] | 15.48 | 21.84 | 10.25 | 13.26 | 18.37 | 4.62 | 5.52 | 5.93 | 1.86 |
| chu *et al.* [24] | 15.97 | 23.66 | 10.80 | 15.41 | 20.21 | 7.06 | 7.64 | 8.69 | 2.52 |
| GPD [7] | 22.87 | 28.53 | 12.84 | 21.33 | 27.83 | 9.64 | 8.24 | 8.89 | 2.67 |
| PointnetGPD [9] | 25.96 | 33.01 | 15.37 | 22.68 | 29.15 | 10.76 | 9.23 | 9.89 | 2.74 |
| GraspNet-baseline [8] | 27.56 | 33.43 | 16.95 | 26.11 | 34.18 | 14.23 | 10.55 | 11.25 | 3.98 |
| gou *et al.* [19] | 27.98 | 33.47 | 17.75 | 27.23 | 36.34 | 15.60 | 12.25 | 12.45 | 5.62 |
| li *et al.* [25] | 36.55 | 47.22 | 19.24 | 28.36 | 36.11 | 10.85 | 14.01 | 16.56 | 4.82 |
| GSNet [17] | 65.70 | 76.25 | 61.08 | 53.75 | 65.04 | 45.97 | 23.98 | 29.93 | 14.05 |
| GSNet [17] + CD | **67.12** | **78.46** | **60.90** | 54.81 | 66.72 | 46.17 | 24.31 | 30.52 | **14.23** |
| Ours | 58.95 | 68.18 | 54.88 | 52.97 | 63.24 | 46.99 | 22.63 | 28.53 | 12.00 |
| Ours + CD | 63.83 | 74.25 | 58.66 | **58.46** | **70.05** | **51.32** | **24.63** | **31.05** | 12.85 |

Table 2: Comparison on Graspnet-1Billion (RealSense). CD denotes Collision Detection.

Although we focus on scale balanced grasp detection in particular tackling small-scale cases, the proposed approach achieves an AP of 63.83% on seen objects, 58.46% on similar objects and 24.63% on novel objects, which outperforms most counterparts. GSNet [17] shows better performance on seen objects mainly because they design an efficient cascaded graspness model to detect grasps in the position which has more successful grasps in its $SO(3)$ space. However, GSNet tends to incur negative impact on the diversity of grasps to some extent since grasps in low graspness positions are not taken into consideration.

## 4.6 Real-world Evaluation

To validate the generalization ability of the proposed approach, we conduct additional real-world experiments. As shown in Figure 8 (a), the grasping system is built on a 7-DoF Agile Diana-7 robot

arm. 30 objects are randomly chosen from the YCB dataset [26], including 10 objects only with grasps of small scales (GSS) (Fig. 8 (b)) and 20 objects with grasps of diverse scales (GDS) (Fig. 8 (c)).

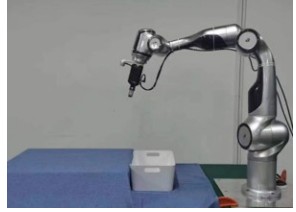 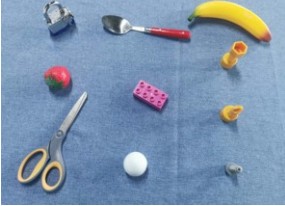 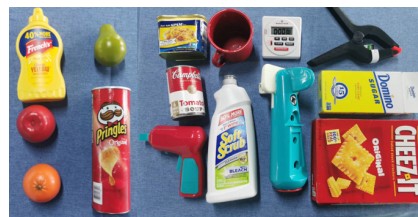

(a) Physical system setting      (b) Objects only with GSS      (c) Objects with GDS

Figure 8: Robot and object settings in the real-world grasping experiments.

We compare our model to the baseline in two settings: isolated object grasping and cluttered object grasping. For isolated object grasping, only the GSS objects are considered and each object is placed in three different poses. Success Rate (SR) is used as the metric. As shown in Table 3, our model delivers a large improvement, which demonstrates its effectiveness in dealing with grasps of small-scales. For cluttered object grasping, 5∼6 objects from both GSS and GDS compose a scene and we make the robot remove them all with a maximum number of operations at 8. SR and Scene Completion Rate (SCR) are employed as the metrics. According to Table 3, our model performs better in both SR and SCR, also indicating its superiority (as the baseline often fails on the GSS objects).

| Model | Isolated | Cluttered | |
| --- | --- | --- | --- |
| | SR (%) | SR (%) | SCR (%) |
| Baseline | 56.67 | 68.42 | 87.10 |
| Ours | 80.00 | 88.24 | 96.77 |

Table 3: Results of the real-world grasping experiments.

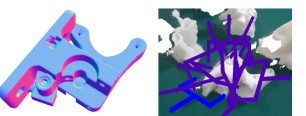 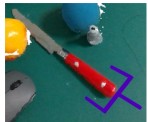

(a) Complicated object      (b) Flat object

Figure 9: Failure cases.

## 5 Limitations

Despite the advantages, our approach still has two limitations. First, as in Fig. 9 (a), our model fails on a very complicated object, which is composed of some small geometrical components. Second, for small-scale flat objects laying in the plane, few grasps are detected, as in Fig. 9 (b) shows. We speculate that the noise is the major reason in the two cases, which requires more discussion on high-precision sensor selection and efficient data refinement.

## 6 Conclusion

In this paper, we work towards scale balanced 6-DoF grasp detection. An MsCG module is proposed to capture comprehensive local description for grasps at different scales. To mitigate the imbalanced scale distribution of grasp labels, we design an SBL loss to adjust the weights of grasp samples in training and an OBS strategy to sample more points on small-scale objects in inference. We also present NcM data augmentation to reduce the influence of sensor noise. Extensive experiments show the advantages of our approach in grasp detection, especially for small-scale samples which are not well handled in the previous work.

**Acknowledgments**

This work is partly supported by the National Natural Science Foundation of China (No. 62022011), the Research Program of State Key Laboratory of Software Development Environment (SKLSDE-2021ZX-04), and the Fundamental Research Funds for the Central Universities.

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
