# OpenReview forum: "Towards Scale Balanced 6-DoF Grasp Detection in Cluttered Scenes"
_robot-learning.org/CoRL/2022/Conference — CoRL 2022 Poster_

### Official Review · Reviewer_iqJc · 2022-07-19

**Originality:** Very Good
**Technical Quality:** Good
**Clarity Of Presentation:** Good
**Impact:** 4

**Recommendation:**

Weak Accept: I recommend accepting the paper, but will not argue for my recommendation if the majority of other reviewers have a different opinion.

**Summary:**

This paper considers the problem of 6DoF robotic grasp detection from point clouds. Small-scale grasp detection is identified as a particular challenge, not having been a focus in prior work. The authors propose a collection of strategies (MsCG, SBL, OBS) geared toward making improvements for the small-scale grasp detection case that is shown to be beneficial on experiments conducted with the GraspNet 1 Billion benchmark.

**Issues:**

Some aspects of the paper are unclear. These are discusses below.

- When is the segmentation network trained? Prior to the main network (point encoder, approach heads and operation head)? If so, the authors could make this more clear to avoid any confusion as I don't see it made explicit in the manuscript. Also, is there a good reason why the same point encoder cannot be used for both the segmentation network as well as the grasp candidate proposal module?

- The loss in equation (2) is not clear without the definition of the terms and sub-losses. While it is a modification of the loss in [8], I suggest to make the definition in equation (2) more clear to make the manuscript self-contained.

- The sentence on line 147-158 that explains how features of new samples are computed is a bit vague. I don't follow how the new feature is the linear interpolation of the three nearest neighbours. Is this the mean of the nearest three? Is there a weight applied to the neighbours based on their distance to the new point?

- The claim on lines 49-51 that point completion and denoising methods are unsuitable is not supported by experiments either conducted in this manuscript or in related work. It would be good to provide quantitative evidence to back up the claim and better justify the design choices of the proposed method.

- In Figure 5, which set of objects are the experiments performed on? Seen, similar, novel or all?

- In Table 2, what does "+ CD" mean?

- Typos
  -- Line 131: "not practical" -> "impractical".
  -- Line 154: "complicate" -> "complicated".

**Quality Of The Limitations Section:**

Limitations are addressed clearly

**Reviewer Expertise:**

4: The reviewer is confident but not absolutely certain that the evaluation is correct

**Robotics Focus:**

Highly relevant to robotics but no hardware experiments

**Strengths And Weaknesses:**

Strengths

The authors address an important problem in the field of robotics and provide a good motivation for the problem they solve; Figure 1 is a good visual example that captures the problem well. The ideas are well conveyed and overall the paper is easy to follow.



Weaknesses

The main weakness for me is that no real robot experiments are performed. While results on the GraspNet 1 Billion benchmark showcase the performance gains of the introduced modules, they are not a substitute for real robot experiments. This is missing and would otherwise make the paper a strong submission.

Secondly, I am not entirely convinced by the contribution of the noisy-clean mix augmentations. The results in Table 1 and Figure 5 reveal that its inclusion does not have a significant impact. We also do not see the performance when the network is trained on purely synthetic scenes, which I think is essential to justify the mixed training data. It would be really interesting to see an ablation on the performance from 100% synthetic (clean) to 100% raw (noisy) to find out what the optimal ratio is.

A minor weakness is that there is no ablation on the number of cylinder radii for the MsCG. Would it help to have more? What are the trade-offs?

**Summary Of Recommendation:**

This paper contains interesting ideas that address a highly relevant in robotics, hence I rate a weak accept given the merit in and contribution of the methodology. Unfortunately, there are key experiments missing (real robot, ablations) that restrict me from giving a strong accept.

---

> ### Author Response · Authors · 2022-08-26
> **Response to Reviewer iqJc (1/2)**
>
> **Comment:**
>
> Dear Reviewer,
>
> Thanks for your positive feedback and valuable comments. We address your concerns below:
>
> ### Q1: Real-world grasping
> Thanks. We conduct additional real-world grasping experiments. The grasping system is built on a 7-DoF Agile Diana-7 robot arm. 30 objects are randomly chosen from the YCB Dataset [1], including 10 objects only with grasps of small scales (GSS) and 20 objects with grasps of diverse scales (GDS) and see Figure 7 in the revised paper for an illustration. We compare our model to the baseline in two settings: isolated object grasping and cluttered object grasping. For isolated object grasping, only the GSS objects are considered and each object is placed in three different poses. Success Rate (SR) is used as the metric. For cluttered object grasping, 5~6 objects of both GSS and GDS compose a scene and we make the robot remove them all with a maximum number of operations at 8. SR and Scene Completion Rate (SCR) are employed as the metrics. The results are shown as follows.
>
> | Model        | Isolated Objects-SR (%) | Cluttered Objects-SR (%) | Cluttered Objects-SCR (%) |
> | ------------ | :---------------------: | :----------------------: | :-----------------------: |
> | **Baseline** |          56.67          |          68.42           |           87.10           |
> | **Ours**     |          80.00          |          88.24           |           96.77           |
>
> For isolated object grasping, our model delivers a large improvement, which demonstrates its effectiveness in dealing with grasps of small scales. For cluttered object grasping, our model performs better in both SR and SCR, also indicating its superiority (as the baseline often fails on the GSS objects).
>
> We add the experimental results of real-world grasping in the revised paper. Besides, a video is attached in the supplementary material.
>
> [1] *Yale-CMU-Berkeley dataset for robotic manipulation research (IJRR2017)*
>
> ### Q2: Contribution of Noisy-clean Mix (NcM)
> For the results in Table 1 in the paper, when NcM is excluded, the mean AP drops 0.89 (41.13 vs. 40.24) and 0.86 (32.67 vs. 31.81) for seen and similar objects respectively. In Table 5 (a), we can also see that NcM significantly improves the success rate for small-scale grasping (ours-full vs. w/o NcM). They both show that NcM helps to achieve scale balanced grasp detection.
> On the other side, we conduct an ablation study on the value of the mix ratio of NcM, where the ratio ranges from 0 to 100% clean data with a step of 25%. The results are shown below (Se = Seen, Si =  Similar, No = Novel).
>
> | Mix Ratio            | Se$_s$ | Se$_m$ | Se$_l$ | Se$_{mean}$ | Si$_s$ | Si$_m$ | Si$_l$ | Si${_{mean}}$ | No$_s$ | No$_m$ | No$_l$ | No$_{mean}$ |
> | :------------------- | :----: | :----: | :----: | :---------: | :----: | :----: | :----: | :-----------: | :----: | :----: | :----: | :---------: |
> | 0% clean (w/o NcM)   | 12.69  | 46.25  | 61.78  |    40.24    |  6.00  | 36.88  | 52.55  |     31.81     |  7.06  | 16.38  | 23.27  |    15.57    |
> | 25% clean (w/ NcM)   | 13.53  | 48.45  | 62.23  |  **41.40**  |  6.95  | 39.72  | 53.81  |   **33.49**   |  7.90  | 17.35  | 23.27  |  **16.17**  |
> | 50% clean (w/ NcM)   | 13.47  | 48.12  | 61.81  |    41.13    |  6.23  | 37.90  | 53.89  |     32.67     |  7.60  | 17.04  | 23.10  |    15.91    |
> | 75% clean (w/ NcM)   | 12.74  | 47.31  | 61.78  |    40.61    |  5.80  | 37.07  | 54.17  |     32.35     |  7.03  | 17.31  | 23.12  |    15.82    |
> | 100% clean (w/o NcM) |  9.62  | 38.87  | 59.86  |    36.12    |  4.10  | 30.23  | 54.41  |     29.58     |  5.17  | 14.67  | 20.64  |    13.49    |
>
> When the NcM module is introduced, it delivers a consistent improvement no matter what the mix ratio is. The value of 25% clean data achieves the best performance and the performance only with clean data (the value is set at 100%) is largely inferior to the others because of the Sim2Real gap.  We add these results in the supplementary material for comprehensive analysis on NcM.
>
> **Zip File:**
>
> /attachment/d7d3c355a5c691f45635653644d75ee9482160c9.zip

---

> > ### Author Response · Authors · 2022-08-26
> > **Response to Reviewer iqJc (2/2)**
> >
> > ### Q3: Ablation on the number of radii for MsCG
> >
> > Thanks for your suggestion. The original MsCG's radius setting is [2cm, 4cm, 6cm, 8cm] and we preliminarily test another two radius settings ([4cm, 8cm] and [2cm, 4cm, 8cm]) for comparison. The results are shown below.
> >
> > | Radii (cm)   | Se$_s$ | Se$_m$ | Se$_l$ | Se$_{mean}$ | Si$_s$ | Si$_m$ | Si$_l$ | Si${_{mean}}$ | No$_s$ | No$_m$ | No$_l$ | No$_{mean}$ |
> > | :----------- | :----: | :----: | :----: | :---------: | :----: | :----: | :----: | :-----------: | :----: | :----: | :----: | :---------: |
> > | [4, 8]       |  9.32  | 44.38  | 59.38  |    37.69    |  2.27  | 34.16  | 53.39  |     29.94     |  4.10  | 14.72  | 21.82  |    13.55    |
> > | [2, 4, 8]    | 13.50  | 46.25  | 60.53  |    40.09    |  6.37  | 36.98  | 52.09  |     31.81     |  7.57  | 15.88  | 21.97  |    15.52    |
> > | [2, 4, 6, 8] | 13.47  | 48.12  | 61.81  |  **41.13**  |  6.23  | 37.90  | 53.89  |   **32.67**   |  7.60  | 17.04  | 23.10  |  **15.91**  |
> >
> > The combination of the features of different radii substantially contributes to the results. When the feature of the radius of 2cm is excluded, the performance of small-scale grasps sharply drops and that of 6cm improves the results of medium- and large-scale grasps. As it takes time to conduct the full ablation on radius, we will complete it and add it in the final version.
> >
> > ### Q4: Segmentation Network
> >
> > Sorry for the ambiguity. The segmentation network is trained prior to the main network. The motivation behind lies in that instance segmentation is a well studied computer vision task and related models can be selected for use in a flexible way. Regarding the architecture with one point encoder for both instance segmentation and grasp detection, we think it can be achieved by carefully tuning the mutual influence of the two sub-tasks. Since it is beyond the current scope of this study and it takes time to well train this model, we would like to leave it in our future work.
> >
> > ### Q5: Incomplete loss definition
> > Thanks for your suggestion. We give the complete definition of each term below. The whole loss consists of the approach loss and rotation loss:
> > $$
> > L = W_i(L^{Approach}({c_i},{s_{ij}}) +\alpha L^{Rotation}(R_{ij},S_{ij},W_{ij}))
> > $$
> >
> > The approach loss is:
> > $$
> > L^{Approach}\left(c_{i},s_{i j}\right)=\frac{1}{N_{c l s}} \sum_{i} L_{c l s}\left(c_{i}, c_{i}^{\*}\right) \\
> > +\lambda_{1} \frac{1}{N_{r e g}} \sum_{i} \sum_{j} c_{i}^{\*} \mathbf{1}\left(\left|v_{i j}, v_{i j}^{\*}\right|<5^{\circ}\right) L_{r e g}\left(s_{i j}, s_{i j}^{\*}\right)
> > $$
> > where $c_i$ is the graspable prediction, $s_{ij}$ is the $j$-th view confidence for point $i$ and $v_{ij}$ is the view direction.
> >
> > The rotation loss is:
> > $$
> > L^{Rotation}(R_{ij}, S_{ij}, W_{ij})= \sum_{d=1}^{K}(\frac{1}{N_{cls}} \sum_{ij} L_{cls}^{d}(R_{ij}, R_{ij}^{\*}) \\ +\lambda_{2} \frac{1}{N_{reg}} \sum_{ij} L_{reg}^{d}(S_{ij}, S_{ij}^{\*})
> >              + \lambda_{3} \frac{1}{N_{reg}} \sum_{ij} L_{reg}^{d}(W_{ij}, W_{ij}^{\*}))
> > $$
> > where $R_{ij},S_{ij},W{ij}$ denote the rotation degrees, grasp confidence scores and gripper widths respectively. $d$ is the depth of the approaching direction.
> >
> > We complete these equations in the revised paper.
> >
> > ### Q6: Feature interpolation
> >
> > The new feature is interpolated by its three nearest neighbors based on their euclidean distances to the given point. The equation is shown below:
> > $$
> > F_{i} = \frac{\frac{F_{n^1}}{D(i,n^1)} + \frac{F_{n^2}}{D(i,n^2)} + \frac{F_{n^3}}{D(i,n^3)}}{\sum_\limits{k=1}^{3}{\frac{1}{D(i,n^k)}}}
> > $$
> > where $F$ is the feature, $D$ is the euclidean distance and $n^k$ is the $k$-th neighbor point.
> > ### Q7: Point-cloud completion and denoising
> > Thanks for your suggestion. point-cloud completion methods mainly work on seen objects because the completion for missing regions of unknown objects is confusing without prior knowledge. A recent paper [1] also gives a similar claim. As for point-cloud denoising, current approaches render noisy data by adding noise in different patterns to clean point-clouds for training models. However, as the distribution of the real noise in existing RGBD sensors is much more complex than that of the pre-defined noise (such as the standard Gaussian noise), it is not easy for such models to generalize. Actually, a SOTA denoising method [2] is preliminarily attempted, but it does not report reasonable results. We rewrite this part with more explanation in the revised paper.
> > [1] *Deep point cloud Reconstruction (ICLR2022)*
> >
> > [2] *Score-Based point cloud Denoising (ICCV2021)*
> >
> > ### Q8: Objects used in Figure5
> > Sorry for the ambiguity. Experiments in Figure 5 are performed on seen objects.
> > ### Q9: The meaning of "+CD"
> > "CD" denotes **C**ollision **D**etection for short. We clarify it in the revised paper.

---

> > > ### Comment · Reviewer_iqJc · 2022-08-28
> > > **Response to authors' answers**
> > >
> > > Thank you to the authors for presenting a strong rebuttal. All my questions have been addressed and the authors have performed the experiments (i.e., real robot as well as ablations on mix ratio and cylinder radii) that were requested. Overall, the paper is much improved as seen in the updated manuscript and supplementary material.

---

### Official Review · Reviewer_XwLV · 2022-07-24

**Originality:** Very Good
**Technical Quality:** Good
**Clarity Of Presentation:** Good
**Impact:** 3

**Recommendation:**

Weak Accept: I recommend accepting the paper, but will not argue for my recommendation if the majority of other reviewers have a different opinion.

**Summary:**

This paper focuses on solving scale imbalance problems in 6-DOF grasp detection tasks. A framework that consists of several modules is proposed. The main contributions include:

1. A Multi-scale Cylinder Grouping (MsCG) module that allows the network to extract features from different scales.
2. A Scale Balanced Learning loss (SBL) to weigh the imbalanced distribution of object sizes in the dataset.
3. A Object balance Sampling (OBS) strategy to deal with the imbalance during inference.

The proposed method is evaluated on the GraspNet-1Billion dataset, and the ablation study shows the effectiveness of each module in the network.

**Issues:**

* What is the method that denotes "baseline" in the evaluation section? It is unclear what method is being compared. It'll be great if it can be explicitly cited.
* In Figure 5, the proposed method without NcM performs better or equally to the full method. Any idea why this is inconsistent with the results in the ablation study?
* What is the "CD" in table 2? I don't see it being mentioned in the paper.

Writings:
* In Figures 1 & 6, how are the good grasp and bad grasp defined?
* In Figure 1 (b), AP is not introduced before. It was confusing for me until I saw the experimental section. I recommend the authors use Average Precision directly instead of an abbreviation.
* In Figure 1 (b), I understand the authors are trying to show with the percentage of small grasps increases, the AP drops accordingly. However, it is also not very clear at the first glimpse. I suggest the authors change the "percentage" to maybe "the percentage of small grasps." in the future.

Nit-Picking:
* I recommend the authors also add color legends in Figure 6. It'll help readers to understand the figure more smoothly.

**Quality Of The Limitations Section:**

Limitations are addressed clearly

**Reviewer Expertise:**

3: The reviewer is fairly confident that the evaluation is correct

**Robotics Focus:**

Highly relevant to robotics but no hardware experiments

**Strengths And Weaknesses:**

**Strengths**

* The proposed method can detect grasps for objects on different scales and is shown to perform better on unseen objects compared to GSNet and other baselines.
* The multi-scale cylinder grouping is an interesting strategy that helps the network to extract features on different scales. It is similar to the image pyramid approach used in the Visual SLAM community.
* The experimental results are thorough and convincing. The ablation study shows the effectiveness of each module. However, some clarification is needed.

**Weaknesses** (Also see Issues)
* The Noisy-clean Mix (NcM) seems to be very similar to the common noise augmentation approach in machine learning. I would appreciate it if the authors could emphasize the difference between the two.
* The paper is generally well-written and easy to follow. However, some of the writings are ambiguous and need to be clarified. (See issues.)

**Summary Of Recommendation:**

This paper proposes a grasp detection framework that addresses the scale-imbalance issue. The results show the proposed method achieves better average precision on unseen objects and the ablation studies of each module are consistent with the claims. There exists some issues that need to be addressed. However, the paper is overall convincing, and thus I recommend a weak accept.

---

> ### Author Response · Authors · 2022-08-26
> **Response to Reviewer XwLV**
>
> **Comment:**
>
> Dear Reviewer,
>
> Thanks for your thorough and positive feedback. We address your concerns below:
>
> ### Q1: Differences between NcM and noise augmentation
>
> Indeed, NcM can be regarded as a noise augmentation method, since more data are generated with different degrees of noise and used in training. However, they work differently in this task.
>
> As we know, synthetic data and raw data represent the shape of a scene in theory and in practice respectively, and they both contribute to training a more powerful model. Unfortunately, they belong to different domains, and the domain gap tends to limit the performance. Noise augmentation adds artificial noise on the synthetic or/and raw samples to enrich the training set. As the artificial noise is quite different from the real noise incurred by the sensor, this method probably enlarges the domain gap, leading to degraded results. In contrary, the proposed NcM module aims to bridge the domain gap by generating more data which randomly mix corresponding synthetic and raw scenes into one sample at instance-level so that the trained model can directly work without fine-tuning.
>
> We clarify this point in the revised paper.
>
> ### Q2: The baseline method
> Sorry for the ambiguity. The baseline method we employ is an enhanced version of graspnet-baseline [1], where a transformer-based point encoder and a new graspable loss with higher threshold are applied for better performance. We clarify it in the revised paper.
> [1] *Graspnet-1billion: A large-scale benchmark for general object grasping. (CVPR2020)*
>
> ### Q3: Inconsistency of the NcM result.
> Sorry for making this confusion. The results in Figure 5 are reached on seen objects. According to Table 1-seen and Figure 5, NcM mainly benefits small- and medium-scale grasping. For small-scale grasping, as illustrated in Section 4.3 (line 206-208), NcM reduces many small-scale grasps but improves the success rate as in Figure 5, since it prevents our network from generating grasps in the region where the shape is seriously corrupted by noise (the AP in Table 1 is improved). For medium-scale grasping, both the number of grasps and success rate are improved in Figure 5, which are also consistent with the results in Table 1.
> ### Q4: The meaning of "CD"
> "CD" denotes **C**ollision **D**etection for short. We clarify it in the revised paper.
> ### Q5: Writings
> Thanks for your valuable suggestions. We make those changes in the revised paper.
>
> **Zip File:**
>
> /attachment/0207db356cd2deab9207cb91d19f4f5f0f12e809.zip

---

> > ### Comment · Reviewer_XwLV · 2022-08-26
> > **Response**
> >
> > Thank you for your response. What you said about NcM makes sense and most of my concerns are addressed.
> >
> > I appreciate the ablation study of NcM provided in the below comments. However, from Figure 5 and the ablation study, it seems NcM only improves the performance slightly (Compare to MsCG), but it introduces the need for a 3D model (point cloud) of the object. I wonder if it will be a good tradeoff to drop the requirement of the 3D model by removing the NcM during training.

---

> > > ### Author Response · Authors · 2022-08-27
> > > **Response to reviewer**
> > >
> > > Thanks for your quick feedback.
> > >
> > > While the effect of NcM is relatively smaller than that of MsCG, NcM still delivers the mean AP improvements of 0.89 (41.13 vs. 40.24) and 0.86 (32.67 vs. 31.81) for seen and similar objects respectively as shown in Table 1. Here, we preliminarily do a Z-test for the results on seen objects and $p=4.47\times10^{-11} < 0.05$, which demonstrates the significance by NcM.
> > >
> > > As for the cost of 3D models, it should be noted that there has been an emerging trend that grasp candidates are sampled by simulation on 3D object models for building benchmarks [1, 2, 3, 4], since it bypasses the high consumption of time and labor required by human annotation [5, 6]. Considering the availability of such 3D models in recent databases, NcM does not incur an extra cost but leads to performance gains.
> > >
> > >
> > >
> > > [1] *Dex-Net 2.0: Deep Learning to Plan Robust Grasps with Synthetic Point Clouds and Analytic Grasp Metrics (RSS2017)*
> > >
> > > [2] *Jacquard: A large scale dataset for robotic grasp detection (IROS2018)*
> > >
> > > [3] *Graspnet-1billion: A large-scale benchmark for general object grasping (CVPR2020)*
> > >
> > > [4] *Egad! an evolved grasping analysis dataset for diversity and reproducibility in robotic manipulation (RA-L2020)*
> > >
> > > [5] *Efficient grasping from rgbd images: Learning using a new rectangle representation (ICRA2011)*
> > >
> > > [6] *Visual manipulation relationship network for autonomous robotics (Humanoids2018)*

---

### Official Review · Reviewer_dLNy · 2022-07-30

**Originality:** Fair
**Technical Quality:** Good
**Clarity Of Presentation:** Good
**Impact:** 3

**Recommendation:**

Weak Accept: I recommend accepting the paper, but will not argue for my recommendation if the majority of other reviewers have a different opinion.

**Summary:**

This paper suggests that most grasps datasets are biased towards large objects than small objects, and as a result grasp planners compute more grasps on large objects/object-parts and struggle to compute grasps on small objects. To address this, the paper proposes the MsCG method to process local features of a point cloud in varying scales, the SBL loss to fix the imbalance using weights during training, the OBS sampling strategy to fix the imbalance during test time, and a data augmentation procedure that mixes raw data with synthetic data.

**Issues:**

The weaknesses section.

**Quality Of The Limitations Section:**

Additional details required

**Reviewer Expertise:**

4: The reviewer is confident but not absolutely certain that the evaluation is correct

**Robotics Focus:**

Highly relevant to robotics but no hardware experiments

**Strengths And Weaknesses:**

Strengths:
1. The problem of data imbalance for 6-dof grasping is an interesting problem, and is well demonstrated in Fig 1 (b).

2. The paper presents a large amount of experiments in simulation with multiple ablations and the results are quite interesting.

3. The MsCG method is interesting and seems to work well empirically.

Weaknesses:
1. The second contribution the paper makes includes weights that scale the training loss according to the proportion of the sample’s class in the dataset, however this seems like a trivial way to address data imbalance and I’d appreciate additional information that supports the novelty of these weights, for example contrasting it with the commonly used method.

2. The third contribution suggests a data augmentation procedure that mixes raw and synthetic data to increase the number of the number of training examples and address the sim2real gap at once. The paper would benefit from additional experiments that compare the performance of NcM with other common data augmentation techniques, for example training on synthetic data and fine-tuning on raw data.

3. The results in tables 1 and 2 seem impressive, however it is not clear what their scale represents, and it might be clearer if the results are normalized in a way that makes the scale more intuitive (% for example).

4. The method includes a step in which “M candidates are sampled…”; What are these candidates? Are these points from the point cloud or grasps, and if grasps how are they represented? Where does the farthest point sampling happen in Fig 2? The input to the point encoder seems to be the entire point cloud, but the output is sampled with dimensions MxC.

5. Was this method evaluated on a physical system? It is not clear that it was, and if not it is strongly recommended to test it.

**Summary Of Recommendation:**

The paper addresses an interesting problem and I’m recommending a weak accept.

---

> ### Author Response · Authors · 2022-08-26
> **Response to Reviewer dLNy (1/2)**
>
> **Comment:**
>
> Dear Reviewer,
>
> Thanks for your valuable feedback. We appreciate your acknowledgment of our idea on the scale imbalance and MsCG modules. We address your concerns below:
>
> ### Q1: Novelty on the Scale Balanced Loss (SBL)
> For the scale imbalanced issue, we tackle it by adopting a loss function to facilitate learning across different grasp sizes. While the loss itself shares the same form as the common ones used in class imbalanced learning, it aims to describe the scale distribution of grasps in scene, where effective weights for grasps of different scales are calculated in a simple manner. More importantly, to the best of our knowledge, we are the first to analyze the scale imbalance problem in grasp learning and introduce the loss function to ease it.
> ### Q2: Comparison of NcM and other data augmentation methods
>
>
> Here, we would like to recall our motivation. Indeed, a straightforward way to mitigate the Sim2Real gap is to conduct a two-phase procedure, i.e. Clean-Train and Noisy-Finetune (CTNF), where the model is firstly trained on synthetic data and then fine-tuned on raw data. Although effective, CTNF incurs additional problems for expected results, including specifically designing the training strategy (e.g. freezing some layers or training different parts of the network with individual learning rates) and carefully setting the relevant hyper-parameters (e.g. training iterations and learning rate). In contrary, the proposed NcM module aims to bridge the domain gap by generating more data which mix synthetic and raw scenes into one sample at instance-level so that the trained model can directly work without fine-tuning.
>
> According to the suggestion, we give a comparison between NcM and CTNF. In CTNF, we train our model using the synthetic data for 18 epochs (the same as in NcM) and finetune it using the raw data for another 12 epochs, with good convergence achieved at both the phases. During fine-tuning, all of the layers of the model are adjusted by a small learning rate (1/10 to that used in training). The results are shown below. (Se = Seen, Si =  Similar, No = Novel)
>
>
> | Model           | Se$_s$ | Se$_m$ | Se$_l$ | Se$_{mean}$ | Si$_s$ | Si$_m$ | Si$_l$ | Si${_{mean}}$ | No$_s$ | No$_m$ | No$_l$ | No$_{mean}$ |
> | :-------------- | :----: | :----: | :----: | :---------: | :----: | :----: | :----: | :-----------: | :----: | :----: | :----: | :---------: |
> | **Noisy-Train** | 12.69  | 46.25  | 61.78  |    40.24    |  6.00  | 36.88  | 52.55  |     31.81     |  7.06  | 16.38  | 23.27  |    15.57    |
> | **NcM**         | 13.47  | 48.12  | 61.81  |  **41.13**  |  6.23  | 37.90  | 53.89  |   **32.67**   |  7.60  | 17.04  | 23.10  |    15.91    |
> | **CTNF**        | 14.87  | 44.75  | 61.60  |    40.41    |  6.66  | 36.99  | 53.60  |     32.42     |  7.75  | 16.95  | 23.44  |  **16.05**  |
>
> In the experiments, we can see that NcM works comparably with CTNF (performs better on the whole) but in a more efficient manner. The inferiority of CTNF is mainly caused by the catastrophic forgetting due to the large gap between the two domains. We add these results in the supplementary material for comprehensive analysis on NcM.
>
> ### Q3: The representation of grasp scales
>
> In Section 4.1, we give the definition of small-, medium- and large-scale grasping, where their grasp widths are in 0-4cm, 4-7cm and 7-10cm, respectively.
> ### Q4: Issues on the network
> Sorry for the ambiguity. The $M$ candidates are the points sampled by Farthest Point Sampling (FPS) from the point-cloud. FPS operates in each set abstraction layer of the point encoder and because we use a two-layer point encoder, FPS is launched two times.
>
> **Zip File:**
>
> /attachment/5ca6f6765d167bea9225295ac05e381cfa923b4c.zip

---

> > ### Author Response · Authors · 2022-08-26
> > **Response to Reviewer dLNy (2/2)**
> >
> > ### Q5: Real-world grasping
> > Thanks. We conduct additional real-world grasping experiments. The grasping system is built on a 7-DoF Agile Diana-7 robot arm. 30 objects are randomly chosen from the YCB Dataset [1], including 10 objects only with grasps of small scales (GSS) and 20 objects with grasps of diverse scales (GDS), and see Figure 7 in the revised paper for an illustration. We compare our model to the baseline in two settings: isolated object grasping and cluttered object grasping. For isolated object grasping, only the GSS objects are considered and each object is placed in three different poses. Success Rate (SR) is used as the metric. For cluttered object grasping, 5~6 objects of both GSS and GDS compose a scene and we make the robot remove them all with a maximum number of operations at 8. SR and Scene Completion Rate (SCR) are employed as the metrics. The results are shown as follows.
> >
> >
> > | Model        | Isolated Objects-SR (%) | Cluttered Objects-SR (%) | Cluttered Objects-SCR (%) |
> > | ------------ | :---------------------: | :----------------------: | :-----------------------: |
> > | **Baseline** |          56.67          |          68.42           |           87.10           |
> > | **Ours**     |          80.00          |          88.24           |           96.77           |
> >
> > For isolated object grasping, our model delivers a large improvement, which demonstrates its effectiveness in dealing with grasps of small scales. For cluttered object grasping, our model performs better in both SR and SCR, also indicating its superiority (as the baseline often fails on the GSS objects).
> >
> > We add the experimental results of real-world grasping in the revised paper. Besides, a video is attached in the supplementary material.
> >
> > [1] *Yale-CMU-Berkeley dataset for robotic manipulation research (IJRR2017)*

---

> ### Comment · Reviewer_dLNy · 2022-08-26
> **Response to authors**
>
> I thank the authors for their in-depth replies and clarifications and I appreciate the additional results from real-world experiments they provide.

---

### Meta-Review · Area_Chair_V3VV · 2022-08-09

**Recommendation:** Accept (Poster)
**Confidence:** 5

**Metareview:**

The paper introduces a new method for 6D grasp detection from point clouds that can deal with small-scale objects. The novelty of the method is on designing feature extraction and training sampling strategies that can handle small-scale objects in 6D grasp detection. The experiments on the GraspNet-1Billion dataset illustrates the effectiveness of the proposed method.

There are some concerns from the reviewers about the Noisy-clean Mix (NcM) strategy in the method. In addition, the paper lacks real-world robot grasping experiments to demonstrate its ability in real robots.

The concerns from the reviewers have been successfully addressed during the rebuttal. The authors are encouraged to revise the final paper accordingly.


**Best Paper Nomination:**

No

---

> ### Author Response · Authors · 2022-08-26
> **Response to Meta-Reviewer**
>
> **Comment:**
>
> Dear meta-reviewer,
>
> Further to the comments and suggestions of the three reviewers, we try our best to address all their comments and concerns in the response and update the paper and supplementary material. Additional experiments are carried out and the modifications are thoroughly explained in the answers to their questions accordingly. These revisions can be summarized as follows:
>
> 1. Following the comments of Reviewer dLNy, XwLV and iqJc, we give more illustration of the **Noisy-clean Mix** (NcM) module with additional ablative and comparative experiments for comprehensive analysis.
> 2. Following the comments of  Reviewer dLNy and iqJc, we add the results of the **real-world grasping experiments**. They are shown in the revised paper and supplementary material (a video is included).
> 3. Following the comments of Reviewer dLNy, XwLV and iqJc, we improve the manuscript where is not so clear, including the literature review, the proposed method, and the experimental setting. Besides, we proofread the revised manuscript and correct typos, grammatical errors, etc.
>
> **Zip File:**
>
> /attachment/b4a9ee33af46748d56016012eccac8c5e28101ae.zip